# Comprehensive Analysis of Rodent-Specific *Probasin* Gene Reveals Its Evolutionary Origin in Pseudoautosomal Region and Provides Novel Insights into Rodent Phylogeny

**DOI:** 10.3390/biology14030239

**Published:** 2025-02-27

**Authors:** Stephan Maxeiner, Lukas Walter, Samuel Luca Zeitz, Gabriela Krasteva-Christ

**Affiliations:** 1Anatomy and Cell Biology, Saarland University, Kirrbergerstr. 100, Bldg. 61, 66424 Homburg, Germanygabriela.krasteva-christ@uks.eu (G.K.-C.); 2Center for Gender-Specific Biology and Medicine (CGMB), Saarland University, 66424 Homburg, Germany

**Keywords:** pseudoautosomal region, rodents, Cricetidae, Muridae, Spalacidae, lipocalin, Muroidea, Eumuroida

## Abstract

The house mouse, *Mus musculus*, is the preeminent rodent employed in basic biomedical research to understand the systemic functions of individual genes in vivo. Genes involved in human pathologies regularly find their counterparts in the mouse genome; however, little is known about genes exclusive to primates or rodents. In this study, we aimed at finding the evolutionary origin of the *probasin* gene, which is only described in mice and rats and is absent in humans. We traced its origin back to when ancestors of mouse-like rodents such as hamsters, mice, and moles separated, consequently becoming absent outside the mammal order of rodents. *Probasin* initially emerged as a gene present on both sex chromosomes in a specialized region that behaves like autosomes, hence termed pseudoautosomal. During the evolution of mouse-like rodents, it recombined to the X-specific region on the X-chromosome. Our study highlights this transition, suggesting that *probasin* has remained, at present, largely conserved in mouse-like species. Close analysis of sequence variations in the *probasin* gene cause us to challenge the current understanding about the phylogenetic relationship among some tribes within the family of mice.

## 1. Introduction

Sex chromosomes, X- and Y-chromosomes, in Eutherian mammals display a subtelomeric region that behaves like autosomes, e.g., by allowing cross-over events during male meiosis, and is therefore named the ‘pseudoautosomal region’ or PAR [1]. Recently, we demonstrated that in mouse-like rodents (superfamily Muroidea), the arrangement of genes within the PAR has undergone dramatic changes, whereas among most mammals, it is largely conserved [2]. The PAR has shrunk in size, accumulated repetitive sequences, and displayed an increase in its overall G and C nucleotide content. Some genes have even translocated to autosomes or been lost [2,3,4]. This loss of genes is intriguing since many of them have been demonstrated to be clinically significant to humans; among these are, e.g., *SHOX* (SHOX homeobox) and *ANOS1* (anosmin 1), responsible for the phenotypes of Turner syndrome or Kallmann syndrome, respectively [2].

Surprisingly, even though both the human and mouse genomes were sequenced and published over two decades ago, a comprehensive analysis of genes present in either the human or the mouse genome has not yet been conducted [5,6]. Current biomedical research highly depends on small rodents; in particular, the accessibility of the mouse genome for genetic manipulation is the avenue to take to understand the role of genes and their products in a systemic in vivo context. Following the fate of PAR genes during rodent evolution, we also observed that while there has been a loss of some genes, novel genes have also appeared. These genes require to be studied given that rodent species roughly comprise about 40% of all mammal species [7].

Recently, we came across *probasin* in a mouse (*Mus musculus*) in *Pbsn*, a gene in the vicinity of *Prkx* (protein kinase cAMP-dependent X-linked catalytic subunit) and *Tbl1x* (transducing beta-like 1 X-linked) localized on the X-chromosome. Both genes are frequently found in or close to the PAR in many mammals [8]. Presumptive ‘*probasin*’ or ‘*probasin*-like’ genes have only been identified in rodent genomes (cf. NCBI databases). Probasin was initially purified as an androgen-dependent basic protein expressed in rat prostate epithelium almost four decades ago and later identified in mice [8,9,10]. It displays a structural resemblance to lipocalins, a protein family present in all kingdoms of life, which include major urinary proteins (MUPs) and odorant-binding proteins (OBPs) in vertebrates; both families are responsible for shuttle lipophilic biomolecules [9,11,12]. Thus far, *Pbsn* knockout in mice has not been generated, leaving the physiological role of probasin protein largely unaddressed. However, its promoter region has been studied thoroughly in rats [9]. For example, androgen-responsive elements of the rat *probasin* promoter have been employed to generate transgenic mice which express oncogenes to study tumor formation in prostate cancer mouse models [13,14]. In a similar approach, transgenic mice expressing Cre-recombinase under the control of a rat *probasin* minimal promoter resulted in *lacZ* reporter gene expression postnatally and largely limited to the prostate epithelium [15].

In this study, we analyzed over sixty rodent genomes within the superfamily Muroidea in which *probasin* genes have been identified. We characterized its respective gene size, base content, and phylogeny. Our results confirm its close relationship with the family of lipocalins and MUPs/OBPs. Additionally, we determined differences in *Pbsn*/*PBSN* sequences, challenging the established relationship of distinct tribes within the subfamily Murinae.

## 2. Materials and Methods

### 2.1. Bioinformatical Analyses

Sequence analysis was essentially performed as has been reported recently [16]. Initially, the sequence information from the house mouse (Gene ID: 54192, NCBI, Bethesda, MD, USA) was used to fetch homologous sequences from rodent genomes available from rodent assemblies curated by Genbank (NCBI) by submitting this sequence to the Blastn suite (https://blast.ncbi.nlm.nih.gov/Blast.cgi?PROGRAM=blastn&PAGE_TYPE=BlastSearch&LINK_LOC=blasthome (accessed on 23 November 2024)) with default settings and either selecting ‘blast’ or ‘megablast’ search options. All sequences were compared using the MultAlin tool (http://multalin.toulouse.inra.fr/multalin/ (accessed on 23 November 2024)) [16]. Once the nucleotide sequence was retrieved for a given species, the exons were identified by their framing splice acceptor and donor sites; the coding region up to the presumptive poly-adenylation signal was isolated and submitted to the ‘Translate tool’ (https://web.expasy.org/translate/ (accessed on 23 November 2024)) to generate the corresponding protein sequence. The newly assembled sequences were used alternatively for further blastn queries. The respective gene location of deduced sequences representing the presumptive start (ATG) of each coding sequence up to the poly-adenylation motif (AATAAA) and the protein sequences can be found in the Appendix A. The genomic region of 61 *PBSN* genes could be fully analyzed (Table 1) regarding exon/intron structure and coding sequences (see Appendix A), consequently 61 coding sequences could be assessed for base composition (cf. Appendix A) and translated into amino acid sequences. Only 42 of the 61 analyzed genes displayed no sequence disruption, hence allowing us to determine the full gene size. The content of the bases C and G either in the full-length coding sequence or in the third position (GC3) was assessed using the GC Content Calculator (https://jamiemcgowan.ie/bioinf/gc_content.html (accessed on 23 November 2024)) and Codon Usage Calculator (https://jamiemcgowan.ie/bioinf/codon_usage.html (accessed on 23 November 2024)) by Jamie McGowan (https://jamiemcgowan.ie/ (accessed on 23 November 2024)). The results can be found in the Appendix A.

Once full nucleotide sequences of the coding regions were identified, they were aligned using Clustal Omega (https://www.ebi.ac.uk/jdispatcher/msa/clustalo (accessed on 23 November 2024)) in the Pearson/FASTA output format. The alignment in the FASTA format was saved and processed using Mega X [17,18]. Sequences were aligned, they were saved in the alignment format, and a phylogenetic tree was produced with default settings using the Construct/Test Neighbor-Joining Tree option. To display the relationship and evolutionary branching points of individual rodent species, a text file listing all analyzed species was uploaded using the Time Tree option of Mega X (Appendix A). To assess genomic changes during the evolution of *probasin*, information available on annotated genomes was retrieved from the NCBI (https://www.ncbi.nlm.nih.gov/gene/?term=probasin (accessed on 23 November 2024)). Information regarding the structure of mouse probasin protein was derived from its entry in the AlphaFold protein structure database (https://alphafold.ebi.ac.uk/entry/O08976 (accessed on 23 November 2024)). In an alignment of selected probasin protein sequences from different Muroidea lineages, the presence of β-sheets and α-helices were highlighted using their respective positions in the mouse. Furthermore, presumptive signal peptides of protein sequences were determined using SignalP-5.0 (https://services.healthtech.dtu.dk/services/SignalP-5.0/ (accessed on 23 November 2024)) [19]. 

### 2.2. RT-PCR Analysis

Prostate tissue was salvaged from one male mouse which had been euthanized by an overdose of isoflurane inhalation (Piramal Critical Care Deutschland GmbH, Hallbergmoos, Germany) followed by cervical dislocation under a license regarding animal procedures for organ collection (see Institutional Review Board Statement) to meet the 3R principles. Prostate tissue was collected and homogenized in TRI Reagent^®^ (Zymo Research Europe GmbH, Freiburg, Germany). Subsequently, the tissue was homogenized by employing a FastPrep-24 5G bead-beating lysis system (MP Biomedicals Germany GmbH, Eschwege, Germany). RNA was isolated using the Direct-zol^TM^ RNA MiniPrep Plus Kit according to the manufacturer’s protocol (Zymo Research Europe GmbH), and RNA concentration was determined using a NanoDrop One spectrophotometer (Thermo Scientific, Waltham, MA, USA). We diluted 1 µg of total RNA to a total volume of 8 µL with RNase/DNase-free water, which was further subject to cDNA synthesis essentially as has been reported recently [20]. Briefly, the RNA was treated with DNase to remove traces of DNA potentially co-purified during the RNA extraction. The DNase-treated RNA was further transcribed into cDNA using the SuperScript II reverse transcription kit (Thermo Scientific). The cDNA was diluted in a 20 µL of RNase/DNase-free water and PCR-reactions were set up using the Q5 High-Fidelity 2× Master Mix according to the manufacturer’s protocol (New England Biolabs GmbH, Frankfurt, Germany). Oligonucleotide binding sites were identified by employing IDT’s Realtime PCR Tool purchased from IDT (Coralville, IA, USA): KC2358 ACGAGTGGCTGGAGTTTTG; KC2359 GAAAATTCCTGATGTCATCTG; SMrt43 GAATAAGGAGGAGATGACGGAG; and SMrt44 GTTGAGTCACTAAGGTTTGATCTTG. PCRs were loaded on a T100 Thermal Cycler (Bio-Rad Laboratories, München, Germany). The PCR temperature settings were as follows: initial denaturation at 98 °C (30 s), 35 cycles at 98 °C (10 s), 60 °C (20 s), 72 °C (30 s), and final extension at 72 °C (1 min). The PCR products were separated on 2% agarose gel supplemented with MIDORI Green (1:10,000; Nippon Genetics Europe GmbH, Düren, Germany) and documented using the ChemiDoc XRS+ System (Bio-Rad Laboratories). Animal research followed the ARRIVE guidelines (where applicable), and animal handling followed the German guidelines for the care and use of laboratory animals approved by local authorities.

### 2.3. Statistical Analyses

Statistical significance was only assessed for the families Muridae and Cricetidae since their sample sizes exceeded n > 3. A normal distribution of data was confirmed by applying the Kolmogorov–Smirnov analysis before a paired Student’s *t*-test was applied. *p*-values below 0.05 were considered significant. Statistical analysis was performed using GraphPad Prism 9 software (GraphPad Software, Boston, MA, USA).

## 3. Results

### 3.1. PBSN Coding Sequences Vary Among Muroidea Families

Thus far, probasin proteins and their underlying genes have only been studied in rats and mice due to their absence in other mammals including humans. In our phylogenetic analysis, we used the mouse *Pbsn* gene and its coding sequence as a reference to investigate its potential origin within the rodent lineage. Therefore, we employed a strategy that we most recently described in detail [16]. At the beginning of our study, the search for *probasin* or *probasin*-like genes retrieved only 36 hits in the NCBI/gene database. We extended this search by submitting the mouse or rat probasin coding sequences to the NCBI/blastn, which resulted in the retrieval of 61 full-length probasin coding sequences. The applied strategy is depicted as a flowchart in Figure 1. The mouse *Pbsn* gene (Gene ID: 54192), or *PBSN* in other rodent species, covered approximately 15.5 kb (distance covering the start codon to the poly-adenylation motif) and consists of seven exons (Figure 2A) [9]. The coding region started on exon 1 and terminated at a stop codon on exon 6, consequently leaving only the 3′ untranslated region, 3′UTR, on exon 7. Similarity searches in the Muroidea family Platacanthomyidae as well as in other rodent families were negative [21]. When we compared the exon/intron structure of all the different rodent species, we noticed that the coding region in rats (family: Muridae), *Peromyscus* spec. (Cricetidae), and *Cricetomys* spec. (Nesomyidae) terminated at the beginning of exon 7 and that their exon 6 was consistently 24 nucleotides shorter than that in *M. musculus* (Figure 2B).

Similarly to mice, the only two available Spalacidae sequences (*N. galili*, *R. pruniosus*) had a preliminary stop codon on exon 6. Generally, exon 2 to exon 5 displayed identical sizes in all analyzed species, i.e., 140, 68, 111, and 102 bases, respectively. The only exceptions were found in the family Spalacidae (cf. Appendix A). A PCR analysis of mouse prostate cDNA using primer sets annealing to different exons revealed that exon 7 was indeed incorporated into the mRNA despite that it did not account for the coding region (Figure 2C).

A more detailed analysis based on alignments of sequences from representative Muroidea species revealed that in *M. musculus*, *Apodemus* spec., and *Praomys delectorum* (Figure 3A and Figure 4), an insertion of 24 bases occurred into exon 6, introducing a preliminary stop codon. However, the splice donor site (‘GT’ nucleotides) following exon 6 and the splice acceptor site (‘AG’ nucleotides) of exon 7 carrying the ‘designated’ stop codon remained largely conserved (Figure 3A,B).

However, there are exceptions in the species *M. musculus*, *M. caroli*, *M. spicilegus*, and *M. spretus*, which all belong to the subgenus ‘*Mus*’ within the genus ‘*Mus*’. A mutation from an A to G nucleotide generated a novel splice acceptor site eight bases upstream of exon 7 (‘AA’ to ‘AG’), which was confirmed by comparing the differences in the exon 6/7 junctions with deposited sequence information of the house mouse (NCBI’s nucleotide repository; AF005204.1, Johnson and Greenberg, see: https://www.ncbi.nlm.nih.gov/nuccore/AF005204.1 (accessed on 23 November 2024)) and rat (M27156.1) [23] (Figure 3C).

### 3.2. PBSN Gene Sizes and Nucleotide Compositions Are Comparable Among Muroidea Species

Given that the Spalacidae sequences show a few distinguishing features compared to all other Muroidea families, we assessed differences in gene size (Figure 5A), the content of G or C nucleotides in the coding sequence (GC content, see Figure 5B), and the GC content in the third codon position (GC3 content, Figure 5C). Where statistical evaluation was applicable, we found a modest but significant difference in the GC content by comparing Cricetidae (41.43%) to Muridae species (40.81%) (Figure 5B). Any other comparisons showed no significant difference. The GC-to-GC3 relationship in Cricetidae, Muridae, and Nesomyidae species followed a linear relationship. Whereas the Spalacidae species *N. galili* spearheaded this relationship, *R. pruinosus* emerged as an outlier (Figure 5D).

### 3.3. Phylogenetic Analysis of Probasin-Coding Sequences Outgroups the Spalacida Family

The phylogenetic relationship was inferred based on the probasin-coding nucleotide sequence of 61 species (Figure 6). The phylogenetic distance of both Spalacidae sequences from all other sequences was consistent with the Time Tree results (Appendix A). The Cricetidae and Muridae families further separated from each other, while the Nesomyidae species were incorporated in the Muridae sub-branch. The above-mentioned shifts in the exon 7 splice acceptor sites in species of the subgenus *Mus* clustered together, despite that exon 7 sequences were not incorporated in this analysis.

### 3.4. Genomic Recombination Events Affect PBSN Gene Localization on the X-Chromosome

Currently, *probasin* sequences are only partially annotated or assigned to a chromosomal locus. To clarify the origin and evolution of *PBSN* beyond what has been published in mice and rats, we chose to compare the genomic vicinities of *PBSN* to determine whether this has been subject to change during rodent evolution (Figure 7). *PBSN* occurred first in Muroidea families. In *N. galili*, it localizes upstream of *PRKX* and *MXRA5* (matrix remodeling associated 5), two genes that are present in the X-specific part of the X-chromosome immediately proximal to the human PAR. In other mammals, both genes mostly reside within the PAR [1]. In Cricetidae (e.g., *P. leucopus*) and different Murdiae species (rat, mouse, etc.), recombination events lead to a translocation of *PRKX* and *PBSN* downstream of *TBL1X* (Figure 7). Additionally, a gene cluster comprising *PRRG1* (proline-rich and Gla domain 1), *LANCL3* (LanC-like family member 3), and *XK* (X-linked Kx-blood-group antigen, Kell, and VSP 13A binding protein) split up during the separation of Muridae and Cricetidae lineages from Spalacidae, eventually ‘homing’ the recombined *TBL1X*/*PRKX*/*PBSN* gene cluster.

Further changes happened in the orientation of *LANCL3* and *XK* on the X-chromosome along the separation into different Muridae lineages. Since the sequence identities and the genomic vicinities of two ‘*probasin*-like’ genes in the Dipodidae species *J. jaculus* are fundamentally different, they were not included in our analysis.

### 3.5. Probasin Proteins Retain Structural Features of Lipocalin Protein Family

Finally, we compared the protein sequences of species representing different Muroidea branches and inferred the structural information available from mouse probasin protein (Figure 8).

All sequences retain a signaling peptide, one α-helix, and nine β-sheets (A to I). These features are conserved, suggesting their similarity to members of the lipocalin and MUP family [9,12]. The first eight β-sheets form a barrel-like structure necessary for the process of shuttling lipophilic biomolecules. Several other features are conserved as well, e.g., a GxW motif (glycine and tryptophane separated by one amino acid) prior to β-sheet A, a CxxxC motif (two cystines separated by three amino acids) following β-sheet B, and a cysteine at the beginning of β-sheet D forming a disulfide bond with a cysteine close to the C-terminus (Figure 8) [9].

## 4. Discussion

The emergence of the lipocalin-like protein probasin remains elusive given that its presence has yet only been described in rats and mice [8,9], and current genomic annotations are only available for *probasin* or ‘*probasin*-like’ genes within rodents (NCBI). Using the mouse and rat coding sequences as references, we were able to identify *probasin* genes by similarity searches in over sixty rodent genomes, which all belong to the Muroidea superfamily. This dates to a first emergence roughly 44–48 million years ago, before the Spalacidae family separated from the Cricetidae, Muridae, and Nesomyidae families, which were also subsumed by the umbrella term Eumuroida (Appendix A) [30]. The genomic vicinity of the *probasin* gene in the Spalacidae and Eumuroida species suggests that it localizes to the X-chromosome, and that at the branching point of both lineages, further recombination events happened. Whereas *PBSN* is located in *N. galili* in a highly conserved gene cluster that is frequently found in the mammalian PAR, in Eumuroida, *PBSN* as well as its neighboring gene *PRKX* both recombined with *TBL1X* to an X-specific region, an event recently described for other genes within this clade [2,3]. Within the evolution of the Eumuroida species, additional recombination events occurred, but this happened without further disrupting the newly joined *TBL1X*/*PRKX*/*PBSN* gene cluster.

Previously, we reported that the Eumuroida PAR is subject to erosion, i.e., genes translocated to autosomes as well as to X-specific regions on the X-chromosome [2,3]. Illustrated by the evolution of the *neuroligin-4* gene, we demonstrated that gene sizes collapsed, alternatively spliced exons were lost, and the GC/GC3 content in synonymous codons increased substantially [3,4]. In contrast to that, generally, *PBSN* genes were unaffected by the aforementioned changes. The GC and GC3 contents averaged between 40 and 45% and 45 and 55%, respectively, and when plotted together, they were distributed in a linear fashion. This was, for example, not the case for *neuroligin-4*, which had GC/GC3 levels starting at 70% and up to over 90%, eventually exiting the linear GC/GC3 relationship, forming a ‘Supermyomorpha island’ [4]. Our observation suggests that *probasin’s* recombination with its neighboring genes to an X-specific localization on the X-chromosome might have happened evolutionarily earlier before its branching from Spalacidae and Eumuroida, saving *PBSN* from the onset of PAR erosion.

Our analysis aimed at placing mouse *Pbsn* in the broader context of other rodent *PBSN* sequences. Interestingly, based on our results, we conclude that mouse *Pbsn* is an outlier compared to the common *PBSN* gene pattern of other rodents in two specific aspects: (a) an insertion into exon 6 introducing a preliminary stop codon and (b) a shift in the splice acceptor site of exon 7. Both features help to assess the proximity of a phylogenetic relationship among certain rodents. The exon 7 splice acceptor shift affects representative species of the subgenus *Mus* within the genus *Mus*, whereas other *Mus* subgenera, i.e., *Coelomys* and *Nannomys*, are unaffected. Since the entire genus *Mus* harbors its *PBSN* stop codon on exon 6, this shift leaves the gene and its translational product unaffected. In our phylogenetic analysis, the sequence insertion in exon 6 appears—whenever present in Muridae species—of common origin. This conflicts with the ‘Time Tree’ depiction (Appendix A), suggesting that the underlying data regarding different Muridae species rely on different source material and need to be re-evaluated carefully. We found insertions in the Muridae tribes Praomyini (e.g., *P. delectorum*), Murini (*M. musculus*), Apodemini (*A. agarius*), and Hydromyini (*U. caudimaculatus*). However, Arvicanthini species (e.g., *A. niloticus*) were unaffected (Figure 5). The insertion into exon 6 challenges the current separation into different tribe branches within the Murinae subfamily. Analysis of nuclear gene sequences and fossil data sets indicate that Arvicantini separated evolutionarily later. This would imply that at least two independent insertions at the very same position occurred [31,32]. Due to this being unlikely, our results put forward a different scenario in which the Praomyini, Murini, Apodemini, and Hydromyini tribes separated evolutionarily later from Arvicanthini. Discordances between current species trees and gene trees might be due to several reasons, such as horizontal gene transfer, gene duplication, and lineage sorting [33]. Our analysis is based on a single gene which emerged first in the common ancestor of Spalacidae and Eumuroida despite its generally similarity to other genes of the large lipocalin family. Our estimate of Muroidea phylogeny based on probasin is similar to recent rodent phylogenies at the level of relationships among families and subfamilies [31,32,34]. The underlying genes as well as *probasin* in this report have the potential to exhibit differences from the species tree [33]. Moreover, all genes in recombining regions including the X-specific part of the mammalian X-chromosome may represent a mosaic of regions with slightly different gene trees [35]. Despite these potential complexities, we found that the evolutionary history of the *probasin* gene, inferred using Neighbor-Joining, exhibited limited topological discordance with the rodent species tree. Recently, Steppan and Schenk acknowledged, regarding the Muridae phylogeny, that despite testing several genes, not all basal core murine nodes were strongly supported; hence, further geographical and morphological features should be taken into account [34].

Probasin proteins appear largely conserved across all analyzed species, displaying the highest similarity in important structural elements and motifs suggesting that functional aspects are retained. Thus far, the structural relationship to lipocalins indicates a potential role in shuttling lipophilic biomolecules. For instance, in co-purification experiments, heparin-binding growth factor-1 has been identified as interacting with probasin [36]. Whether the role of these biomolecules is exclusive to the prostate remains elusive and needs to be addressed in future experiments. Probasin has a close structural relationship to other lipocalins such as major urinary proteins and odorant-binding proteins, which are both involved in the shuttling of pheromones [12]. Combinations of these proteins are supposed to potentially have a signature role for different species [12,37]. The addition of probasin to this ‘cocktail’, which is produced by the prostate and, hence, added to seminal fluid, might contribute to a signature blend of odorants shuttled by MUPs and OBPs. We need to await the generation of probasin-deficient mice to assess its potential beneficial evolutionary role.

## 5. Conclusions

*Probasin* is a unique lipocalin-like gene found in a subset of mouse-like rodents, Muroidea. It likely emerged in common ancestors within the pseudoautosomal region of both sex chromosomes. However, recombination events during evolution resulted in its current placement in the X-specific region of the X-chromosome in, e.g., the house mouse. Analysis of the coding region suggests that the translocation from the pseudoautosomal region elsewhere occurred prior to changes that affected its coding region, i.e., no increase in the GC or the GC3 content in its coding region is observed. Subtle changes to its coding region such as preliminary stop codons in exon 6 or changes to splice acceptor sites in exon 7 help to clarify evolutionary relationships among closely related species on the infra-family level of Muridae.

## Figures and Tables

**Figure 1 biology-14-00239-f001:**
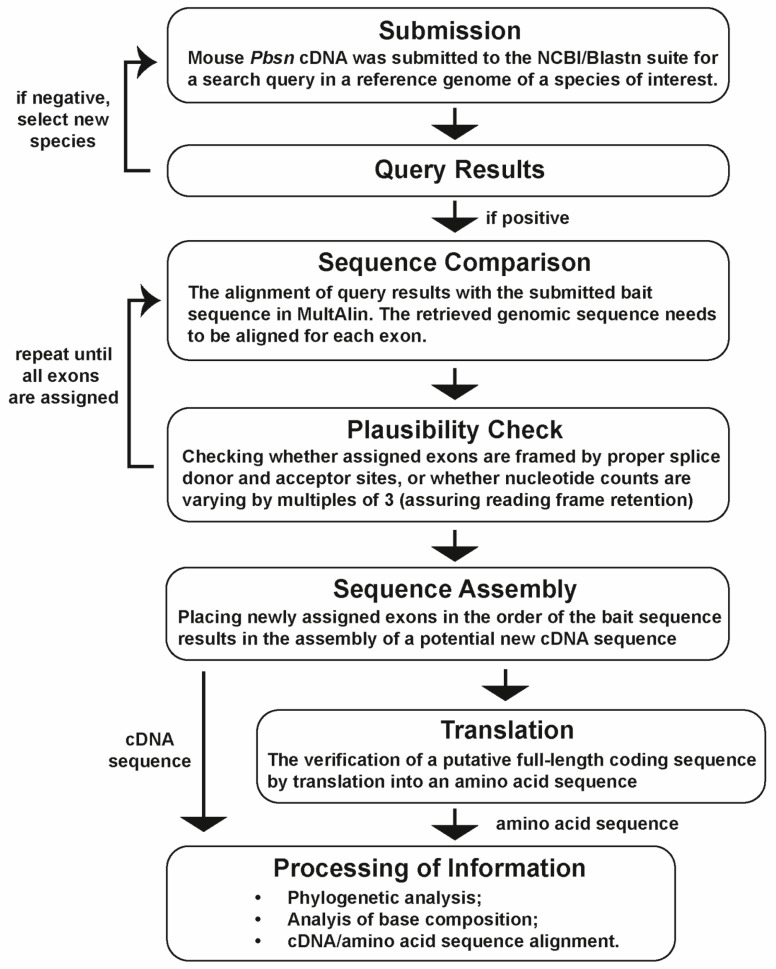
A flowchart depicting each step of the process of identifying novel, yet undescribed, probasin coding sequences.

**Figure 2 biology-14-00239-f002:**
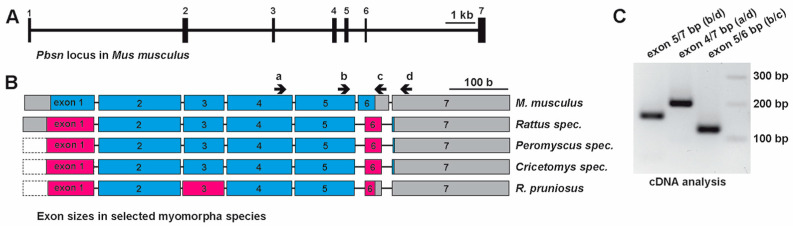
Characterizations of probasin gene features in the rodent superfamily Muroidea. (**A**) A general overview of the *probasin* gene locus in the house mouse, *Mus musculus*. Exons are numbered and their respective positions are drawn to scale. (**B**) A comparison of relative exon sizes (numbered and drawn to scale), coding region differences relative to the house mouse (blue, shared sizes; magenta, deviating sizes), and untranslated regions (gray). Dashed-lined open boxes indicate that the size of the untranslated region is not determined. The house mouse and rats, *Rattus* spec., represent the family Muridae, *Peromyscus* spec. represent Cricetidae (hamsters), *Cricetomys* spec. represent the Nesomyidae family, and the *R. pruniosus* species gene information is derived from a Spalacidae representative. Black arrows with respective lower case letters a to d indicate relative positions of different primers used in the RT-PCR experiment. (**C**) cDNA analysis derived from mouse prostate tissue employing primer combinations localizing to different exons confirms the presence of exon 7 in *Pbsn* transcripts in the house mouse. Abbreviations: b, base; kb, kilobase; base pair.

**Figure 3 biology-14-00239-f003:**
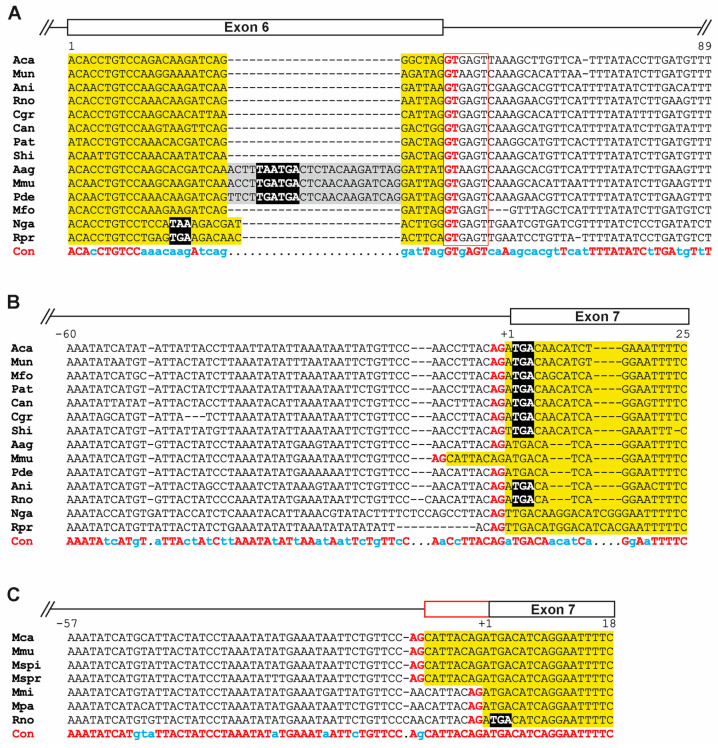
Differences within probasin-coding nucleotide sequences. (**A**) The alignment of selected nucleotide sequences comprising full-length exon 6 and parts of the subsequent intron. (**B**) The alignment of selected nucleotide sequences comprising the beginning of exon 7 as well as parts of the upstream intron sequence. (**C**) The alignment of selected nucleotide sequences from the genus *Mus* and from rats comprising the beginning of exon 7 as well as parts of the upstream intron sequence. Alignment was performed using MultAlin [22] with default settings using the Symbol comparison table ‘DNA-5-0’; in the case of (**B**), Gap penalties at the ‘end’ were chosen. In (**A**–**C**), nucleotide sequences highlighted in yellow represent exon sequences, and gray highlights insertions exclusive to the Murinae species Aag, Mmu, and Pde. The list of abbreviated species can be found in Appendix B. All stop codons are depicted in black with a white font. The consensus (Con) shows conserved nucleotides using a red font and upper case; a blue font with lower case indicates frequently used nucleotides in the alignment. (**A**) The open box in red indicates a splice donor site following the coding region of exon 6. All splice donor (GT) and splice acceptor sites (AG) are depicted in an upper-case red font.

**Figure 4 biology-14-00239-f004:**
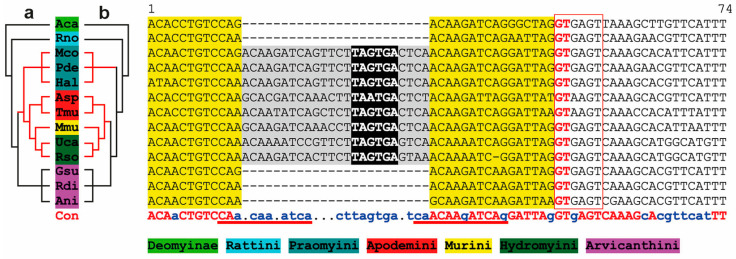
A comparison of sequence insertions into *probasin* exon 6 within the subfamilies Murinae and Deomyinae. Selected sequences of 13 different representative Murinae and Deomyinae species are aligned, comprising full-length exon 6 (yellow) and the downstream intron sequence. Stop codons emerging in the reading frame are highlighted in black with a white font. Insertions of additional sequences in a subset of species are depicted in gray. Alignment was performed using MultAlin with default settings using the Symbol comparison table ‘DNA-5-0’ [22]. A consensus sequence with identical bases (upper case, red) and frequent bases (lower case, blue) is displayed underneath the species’ alignment. Potential sequence duplications are underlined (red). The species are color-coded referring to the respective subfamily or tribe. (**a**) The revised relative phylogenetic relationship based on our findings. (**b**) The relative phylogenetic relationship presented by Aghová and co-workers [2]. Species abbreviations can be found in Appendix B. The open red box indicates splice donor site following the coding region of exon 6. The splice donor (GT) is depicted in an upper-case red font.

**Figure 5 biology-14-00239-f005:**
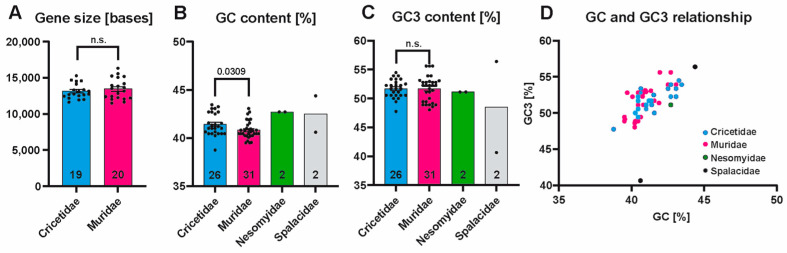
(**A**) A comparison of gene sizes between Cricetidae and Muridae species. (**B**,**C**) Comparisons of the average GC content in the coding sequence (**B**) or in the third position of each codon (**C**). (**D**) An XY scatter blot depicting the GC-and-GC3 relationship in different species. In (**A**–**C**), all columns are labeled with the number (n) of species included, and the sizes represent the mean size/content of analyzed species, and, where applicable, error bars are inserted representing the S.E.M. (standard error of the mean). Abbreviations: GC, guanosine/cytidine nucleotides; GC3, guanosine/cytidine nucleotides in the third position of a codon; n.s., not significant.

**Figure 6 biology-14-00239-f006:**
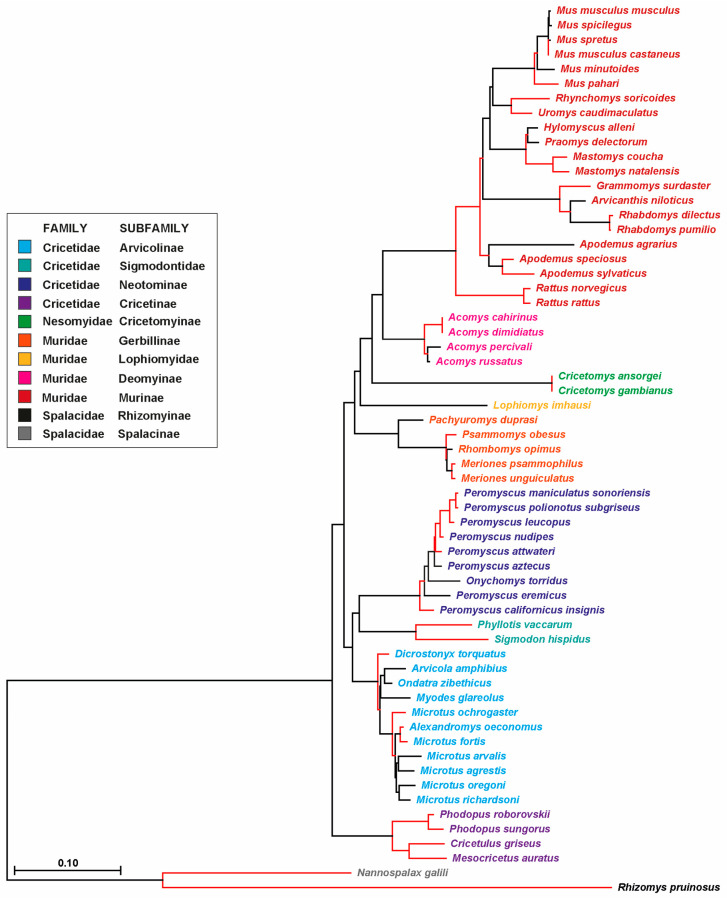
A phylogenetic tree of probasin-coding nucleotide sequences. The evolutionary history was inferred using the Neighbor-Joining method [24]. The optimal tree with the sum of a branch length = 3.11734955 is shown. The percentage (>70) of replicate trees in which the associated taxa clustered together in the bootstrap test (1000 replicates) [25] are indicated by red branches. The tree is drawn to scale, with branch lengths in the same units as those of the evolutionary distances used to produce the phylogenetic tree. The evolutionary distances were computed using the Maximum Composite Likelihood method [26] and are in the units of the number of base substitutions per site. The rate variation among sites was modeled with a gamma distribution (shape parameter = 1). The differences in the composition bias among sequences were considered in evolutionary comparisons [27]. This analysis involved 61 nucleotide sequences. All ambiguous positions were removed for each sequence pair (pairwise deletion option). There was a total of 540 positions in the final data set. Evolutionary analyses were conducted in Mega X [17,18].

**Figure 7 biology-14-00239-f007:**
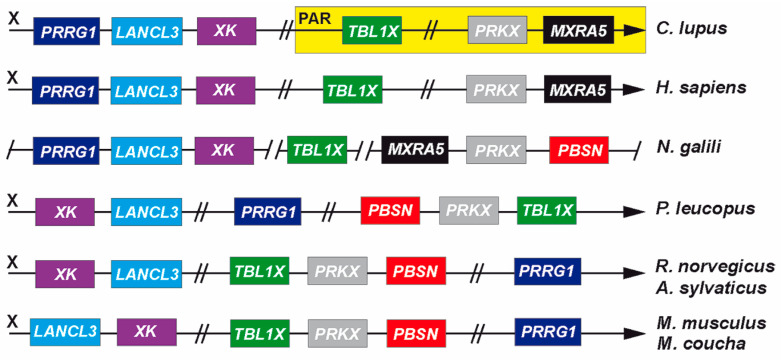
Recombination events affecting *PBSN* gene localization on the X-chromosome. A comparison of the *probasin* gene location relative to its neighboring genes in selected annotated X-chromosomes of the domestic dog (*C. lupus familiaris*), humans (*H. sapiens*), the Cricetidae species *Peromyscus leucopus* (‘white-footed mouse’), and several Muridae species: *Rattus norvegicus*, *Apodemus sylvaticus*, *Mastomys coucha*, and the house mouse. The dog genome is depicted to indicate that the genes *TBL1X*, *PRKX*, and *MXRA5* are located in the pseudoautosomal region (PAR) of the X- and Y-chromosomes (yellow-shaded region), whereas in humans, this genomic vicinity is already incorporated in the X-specific region of the X-chromosome [1]. In the genome of the Spalacidae species *N. galili*, the vicinity of several genes was identified but lacks formal assignment to the X-chromosome. Arrowheads point to the telomer. Double-hyphenated bars indicate unspecified distances between depicted genes. Genes are presented in the order of appearance. The occurrence of recombination events in the evolution of the X-chromosome in Cricetidae and Muridae indicate that *PRKX* and *PBSN* recombined next to *TBL1X*. Additionally, the gene cluster, including *PRRG1*, *LANCL3*, and *XK*, fell apart, now homing among other genes the cluster comprising *TBL1X*, PRKX, and *PBSN*.

**Figure 8 biology-14-00239-f008:**
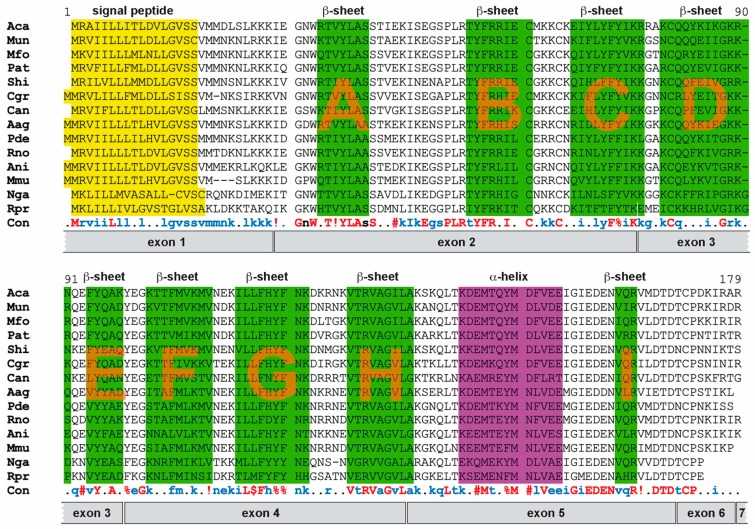
A comparison of probasin protein sequences representing different Muroidea families and subfamilies. Protein sequences were aligned using MultAlin [22], and different protein features are highlighted, imposing the information available for the presumptive structure of mouse probasin selected from the AlphaFold2.0 database [28,29]. Green highlighted regions represent β-sheets, and the lilac region depicts an α-helix. β-sheets are listed alphabetically in their order of appearance. Signal peptides are highlighted in yellow. Exon coverage is depicted in the gray-shaded boxes underneath the translated consensus protein sequence. Amino acids are represented by one-letter code. The color-codes and symbols of the amino acid consensus (Con) are the following: red, identical amino acids; blue, frequent consensus of a specific amino acid; !, either I or V; $, either L or M; %, F or Y; #, N, D, Q, E, B, or Z. Species abbreviations can be found in Appendix B.

**Table 1 biology-14-00239-t001:** Overview of analyzed genomes available from NCBI databases.

Family	Species
Cricetidae	*Alexandromys oeconomus* ^1^, *Arvicola amphibius* ^2^, *Cricetulus griseus* ^2^, *Dicrostonyx torquatus* ^1^, *Mesocricetus auratus* ^1^, *Microtus agrestis* ^1^, *M. arvalis* ^1^, *M. fortis* ^2^, *M. ochrogaster* ^1^, *M. oregoni* ^2^, *M. richardsoni* ^1^, *Myodes glareolus* ^2^, *Ondatra zibethicus* ^1^, *Onychomys torridus* ^2^, *Peromyscus attwateri* ^1^, *P. aztecus* ^1^, *P. californicus insignis* ^2^, *P. emericus* ^2^, *P. leucopus*, ^2^ *P. manniculatus sonoriensis* ^2^, *P. nudipes* ^1^, *P. polionotus subgriseus* ^1^, *Phodopus roborovskii* ^2^, *P. sungorus* ^1^, *Phyllotis vaccarum* ^1^, *Sigmodon hispidus* ^1^
Muridae	*Acomys cahirinus* ^1^, *A. dimidiatus* ^1^, *A. percivali* ^1^, *A. russatus* ^2^, *Apodemus agrarius* ^1^, *A. speciosus* ^1^, *A. sylvaticus* ^2^, *Arvicanthis niloticus* ^2^, *Grammomys surdaster* ^2^, *Hylomyscus alleni* ^1^, *Lophiomys imhaus* ^1^, *Mastomys coucha* ^2^, *M. natalensis* ^1^, *Meriones psammophilus* ^1^, *M. unguiculatus* ^2^, *Mus minutoides* ^1^, *M. musculus* ^2,3^, *M. musculus castaneus* ^1^, *M. pahari* ^2^, *M. spicilegus* ^1^, *M. spretus*^1^, *Pachyuromys duprasi* ^1^, *Praomys delectorum* ^1^, *Psammomys obesus* ^2^, *Rattus norvegicus* ^2^, *R. rattus* ^2^, *Rhabdomys dilectus* ^1^, *R. pumilio* ^1^, *Rhombomys opimus* ^1^, *Rhynchomys soricoides* ^1^, *Uromys caudimaculus* ^1^
Nesomyidae	*Cricetomys ansorgei* ^1^, *C. gambianus* ^1^
Spalacidae	*Nannospalax galili* ^2^, *Rhizomys pruinosus* ^1^

^1^ Genomes without *PBSN* gene annotation. ^2^ Genomes with *PBSN* or *PBSN*-like gene annotation. ^3^ Tissue processing for cDNA analysis. All species are presented in alphabetical order in each family section.

## Data Availability

The relevant data supporting the findings of this study are available in this article and its Appendix A.

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
