# Peer review of "Comprehensive Analysis of Rodent-Specific Probasin Gene Reveals Its Evolutionary Origin in Pseudoautosomal Region and Provides Novel Insights into Rodent Phylogeny"

_biology, 2025, doi:10.3390/biology14030239_

Round 1
Reviewer 1 Report
Comments and Suggestions for Authors
L36 The sentence “Rodents represent the largest mammalian order (~40%)” is not clear. Represent about 40% of known mammalian species.
L28-39 the paragraph must be improved, please rethink on the order of sentences. L36 sentence should be first-second sentence of the paragraph. links between L30-33 sentences should be improved. L33 vice versa should be in italic.
L40 not clear why this sentence is needed here, similar ideas already written in L30-33, also why here “in contrast”
At the end of introduction authors state that they analysed over 60 genomes of rodents, however, in Materials and Methods they start with RT-PCR subsection. It would be very beneficial that at the beginning of Methods authors would describe in detail how animal material was gathered, include table of animal species included in the study; also to describe which species were analysed in current work and what data were gathered from databases.
L64 what do you mean by “one male mouse” how many species? only mice were analysed not other animals of the superfamily Muroidea?
Supplementary Data File. In excel sheets different number of species are presented (61,60,42). Could you please clarify reasons for this.
L123 Figure S2 is cited prior to Figure S1, please correct.
Please correct headings format L141, L252, L270, L317 as it is in Materials and Methods section
L145-147 is must be moved to introduction
L151-155 it should be moved into Methods section. The main drawback of this work, it is not entirely clear what was done in previous studies and what in this one.
L159 what do you mean here by “mice”, Mus musculus, genus Mus or?
I cannot find Figure 2 citing in the text before the figure body.
About the GC content L243-249, however the Figure 1 is in L175-180, it is messy presentation of the data, hard to follow. One of the option splitting Figure 1 into two separate ones.
L244-245 “These features had been subject to change 244 in earlier studies [2,5].” This should be moved to discussion if needed
Figure 4. Please redo. For readers it is inconvenient to look at Figure 4 and Figure S1 to see full data. Figure S1 is redundant. Please insert bootstrap values higher than 70 in Figure 4 or alternatively you can use for example red colour to mark branches that have higher than 70 bootstrap values.
Figure 4. Sigmodon hispidus and Phyllotis vaccarum should be differently coloured than representatives of the genus Peromyscus.
Author Response
We would like to thank Reviewer 1 for his/her/their very constructive criticism of our submitted manuscript. The points that were raised are demonstrating that the article has been read in detail, something that we appreciate very much.
The preliminary manuscript prior to submission was formatted for “Cells”, and upon evaluation redirected to “Biology”. Given the scope of “Biology” we agree that it is a better fit. Since the reviewer has evaluated and commented on the “Cells” version, we would like to point out that we have now transferred the version for resubmission to the “Biology” template. We have indicated all changes to the manuscript by highlighting it with yellow color.
Please, find below the point-to-point reply to the criticism.
We hope that Reviewer 1 finds our corrections and improvements appropriate to eventually granting his/her/their approval for publication in Biology.
We further want to point out that figures were rearranged, and a flowchart and table have been introduced. Some of the changes aim also to meet criticism/comments made by Reviewer 2 and 3.
Thank you and kind regards,
Stephan Maxeiner on behave of all authors
Point-to-point reply
Comment 1: L36 The sentence “Rodents represent the largest mammalian order (~40%)” is not clear. Represent about 40% of known mammalian species.
Response 1: The introduction has been rearranged and rewritten, and the sentence reads now as “that rodents roughly comprise about 40% of all mammal species”
Comment 2: L28-39 the paragraph must be improved, please rethink on the order of sentences. L36 sentence should be first-second sentence of the paragraph. links between L30-33 sentences should be improved. L33 vice versa should be in italic.
Response 2: The introduction was rewritten. We hope that its structure now is more appealing. Please, note during the preparation of this paragraph the final citations haven’t yet been inserted. They are, however, present in the submitted manuscript.
“Sex chromosomes, X- and Y-chromosomes, in Eutherian mammals display a subtelomeric region that behaves like autosomes, e.g., allowing cross-over events during male meiosis, therefore named ‘pseudoautosomal region’ or PAR [1]. Recently, we could demonstrate that in mouse-like rodents (superfamily Muroidea) the among most mammals largely conserved arrangement of genes within the PAR has undergone dramatic changes [2]. The PAR has shrunken in size, accumulated repetitive sequences, and displayed an increase in its overall G and C nucleotide content. Some genes even had translocated to autosomes or had been lost [2-4]. This loss of genes is intriguing since many of them have been demonstrated to be clinically significant to humans, among these are, e.g., SHOX (SHOX homeobox) or ANOS1 (anosmin 1), responsible for the phenotypes of Turner syndrome or Kallmann syndrome, respectively [2].
Surprisingly, even though the human and mouse genome have both been sequenced and published over two decades ago, a comprehensive analysis about genes being present in either the human or the mouse genome still awaits to be complied [5,6]. Current biomedical research highly depends on small rodents, in particular the accessibility of the mouse genome to genetic manipulation is the avenue to take to understand the role of genes and their products in a systemic in vivo context. Following the fate of PAR genes during rodent evolution, we also observed that one does not need to mourn the loss of genes but that also novel genes happen to arise. These genes in their own respect demand to be studied given that rodent species roughly comprise about 40% of all mammal species [7].”
Comment 3: L40 not clear why this sentence is needed here, similar ideas already written in L30-33, also why here “in contrast”
Response 3: We concur and therefore have deleted the sentence.
Comment 4: At the end of introduction authors state that they analyzed over 60 genomes of rodents, however, in Materials and Methods they start with RT-PCR subsection. It would be very beneficial that at the beginning of Methods authors would describe in detail how animal material was gathered, include table of animal species included in the study; also, to describe which species were analyzed in current work and what data were gathered from databases.
Response 4: We have rearranged the order of information presented in the methods section. Now, it starts with the bioinformatical part followed by the RT-PCR part. We incorporated the requested table indicating which information was available from NCBI databases, also indicating that mouse tissue was the only one that we processed. Please, note that mouse information was available before, but that mouse prostate tissue was processed as a proof for the predicted presence of exon 7.
Comment 5: L64 what do you mean by “one male mouse” how many species? only mice were analysed not other animals of the superfamily Muroidea?
Response 5: Given the reported expression of probasin in prostate tissue, we analyzed cDNA derived from this specimen to demonstrate the splicing pattern of probasin. The literature was not clear about it. Our phylogenetic analysis clearly indicates that exon 7 carries the stop codon for the probasin coding sequence, the sequence of the lab mouse, however, carries an upstream stop codon on exon 6. We aimed at clarifying whether exon 7 was still present in probasin cDNA.
All other sequences derived from data available from NCBI’s GenBank. These data were only partially annotated and some of them were falsely annotated by algorithms, deemed “predicted” We applied a strategy recently published by some of the authors (Maxeiner et al., J Physiology, 2023, Pitfalls of using sequence databases for heterologous expression studies – a technical review”). The strategy is now depicted as a flowchart as suggested by Reviewer 2.
Comment 6: Supplementary Data File. In excel sheets different number of species are presented (61,60,42). Could you please clarify reasons for this.
Response 6: We appreciate the keen eye of the reviewer, since it brought a small correction not affecting the manuscript and study itself. Firstly, we noticed that the lane for the species “Mastomys coucha” was missing in our ‘genes & comments’ section of the Supplementary Data Sheet. We added the information resulting in 61 entries. The same number of species are displayed in the ‘GC & GC3’ section of analyzed coding sequences. Secondly, we noticed that in the latter section the name of the root vole, Microtus oeconomus, was used, whereas in the section regarding the gene annotation the more recent name Alexandromys oeconomus was used. We corrected this now in both sections to Alexandromys oeconomus. The information about the gene size could only be used from 42 out of 61 genes. Only 42 genes were fully sequenced from the presumptive start codon to the polyadenylation side. We excluded 19 of the 62 analyzed genes since they were disrupted by stretches of unspecified nucleotides, however not affecting the potential coding sequences.
We added the following sentences to section 2.1. Bioinformatical Analyses “The genomic region of 61 PBSN genes could be fully analyzed (Table 1) regarding exon/intron structure and coding sequences (see Supplementary Data Sheet, section ‘gene structure & comments’), consequently 61 coding sequences could be assessed for base composition (cf. section ‘GC & GC3 content’) and translated into amino acid sequences. Only 42 of 61 analyzed genes displayed no sequence disruption, hence, allowing us to determine the full gene size.”
Comment 7: L123 Figure S2 is cited prior to Figure S1, please correct.
Response 7: Due to omission of Figure S1, Figure S2 replaces Figure S1. The reference in the text has been changed accordingly.
Comment 8: Please correct headings format L141, L252, L270, L317 as it is in Materials and Methods section
Response 8: Given that the style was presented properly (italic, font and font size) we assume that the reviewer refers to the aspect that headings should reflect briefly the result of the following paragraphs. We changed the headings accordingly. Furthermore, a new heading was introduced because of splitting former Figure 1 into two separate figures, now Figure 2 and Figure 5. We hope the reviewer finds this acceptable.
Comment 9: L145-147 is must be moved to introduction
Response 9: During the revision of the introduction, we have incorporated this sentence in the revised introduction.
Comment 10: L151-155 it should be moved into Methods section. The main drawback of this work, it is not entirely clear what was done in previous studies and what in this one.
Response 10: The sentences have been removed from the Results section and incorporated into the revised Methods section.
Comment 11: L159 what do you mean here by “mice”, Mus musculus, genus Mus or?
Response 11: “mice” has been changed to “M. musculus” for clarification.
Comment 12: I cannot find Figure 2 citing in the text before the figure body.
Response 12: The sequence of figures was rearranged. We took care to avoid a mistake like that mentioned by the reviewer
Comment 12: About the GC content L243-249, however the Figure 1 is in L175-180, it is messy presentation of the data, hard to follow. One of the option splitting Figure 1 into two separate ones.
Response 13: We are particularly grateful for pointing out this flaw. We took up the suggestion and have split original Figure 1 into two separate parts. In the flow of the Results section the part regarding gene features is now placed as new Figure 5. The first part of previous Figure 1 is now Figure 2. All subsequent figures have been renamed correspondingly. I think this strengthens the line of though in our presentation of the results.
Comment 14: L244-245 “These features had been subject to change 244 in earlier studies [2,5].” This should be moved to discussion if needed
Response 14: This sentence has been removed. It has been addressed similarly in the Discussion already.
Comment 15: Figure 4. Please redo. For readers it is inconvenient to look at Figure 4 and Figure S1 to see full data. Figure S1 is redundant. Please insert bootstrap values higher than 70 in Figure 4 or alternatively you can use for example red colour to mark branches that have higher than 70 bootstrap values.
Response 15: We appreciate this suggestion. We decided to indicate values over 70 by red branches and therefore deleted Figure S1. Formerly Figure 4 is now Figure 6.
Comment 16: Figure 4. Sigmodon hispidus and Phyllotis vaccarum should be differently coloured than representatives of the genus Peromyscus.
Response 16: We changed the color while changing the branch colors (see above).
Reviewer 2 Report
Comments and Suggestions for Authors
In this article, Stephan et al. propose a study on the rodent specific gene probasin. They mainly discussed the evolutionary origin of probasin gene, its distribution in rodents and its impact on rodent evolution. Before considering publishing in our journal, some revisions are needed.
Comments:
1. Our journal is “Biology”, but the authors use the template of “Cells”. Please revise it.
2. Please divide the bioinformatics analysis to several paragraph to facilitate the understanding
3. A flowchart of the work is need.
4. For the abbreviated species name, only mention once in the first time. In addtion, please add a section of Abbreviation list after the Conclusion section
5. In Figure 6, are the color-codes and symbols of the amino acid consensus common in sequence alignment? In our previous experience of multiple sequence alignment by Clustal, I did not see this symbol.
6. Although it is currently believed that the probasin gene is unique to rodents, further study of the probasin-like genes of other mammals may reveal homologous genes with similar functions or structures to the probasin gene.
7. Please discuss how the probasin gene can affect the evolution of rodent?
8. Only use RT-PCR in bioinformatics analysis were considered simple work. Please find some other data with transcriptomic, proteomic and epigenomic data, to make more comprehensively understand the role and regulation mechanism of probasin gene in different biological processes
Author Response
We would like to thank Reviewer 2 for his/her/their criticism, and the comments raised. Please, find below a point-to-point reply to the comments. We would like to point out, that some of the criticism/comments might already be addressed in context with the response to Reviewer 1 and Reviewer 3.
We hope that Reviewer 2 finds our corrections and improvements appropriate to eventually granting his/her/their approval for publication in Biology.
Thank you and kind regards,
Stephan Maxeiner on behave of all authors
Point-to-point reply
Comments:
- Our journal is “Biology”, but the authors use the template of “Cells”. Please revise it.
Response 1: The manuscript and all changes to it has now been transferred to the “Biology” template. Originally, the manuscript had been submitted to “Cells”. The editor suggested to transfer the manuscript to “Biology” as a better fit. I apologized for any inconvenience/confusion on the side of Reviewer 2.
During the revision process the manuscript has been formatted to the “Biology” template provided by the handling editor.
- Please divide the bioinformatics analysis to several paragraph to facilitate the understanding
Response 2: As suggested by Reviewer 1, we have split the original Figure 1 with inserts A to G into two parts, which are now Figure 1 A to C, and a new Figure 4 with inserts A to D. The new arrangement of the figures now splitting genomic organization of the probasin gene in selected species and analyses regarding its size and nucleotide profile fits better to their representation and appearance in the Results section. We introduced another heading to clearly separate this section “3.2. PBSN gene sizes und nucleotide compositions are comparable among Muroidea species”
- A flowchart of the work is need.
Response 3: A flowchart with figure legend has been added as new Figure 1. The numbering of all other figures has changed accordingly. We believe that this adds clarity and also reflects the comment raised under 2.)
- For the abbreviated species name, only mention once in the first time. In addition, please add a section of Abbreviation list after the Conclusion section
Response 4: We have added section “Appendix A” in which all abbreviations for the species are explained to avoid repetition in what are now Figures 3,4 and 8.
- In Figure 6, are the color-codes and symbols of the amino acid consensus common in sequence alignment? In our previous experience of multiple sequence alignment by Clustal, I did not see this symbol.
Response 5: The colors and symbols in our sequence alignment derive from the interface “MultAlin” from Corpet et al. This has been referenced in all figure legends. Please, refer to the following link:
http://multalin.toulouse.inra.fr/multalin/
- Although it is currently believed that the probasin gene is unique to rodents, further study of the probasin-like genes of other mammals may reveal homologous genes with similar functions or structures to the probasin gene.
Response 6: To date, probasin or probasin-like genes have only been identified in rodents. The current gene entry list on the NCBI website https://www.ncbi.nlm.nih.gov/gene/?term=probasin+or+probasin-like displays 36 hits. No other species outside the rodent order shows any hit. We acknowledge, however, that in the literature the structure of “probasin” has been compared to the large and abundant family of lipocalins. We address this observation in the third paragraph of the introduction with the following sentence: “It displays structural resemblance to lipocalins, a protein family present in all kingdoms of life, and major urinary proteins (MUPs) found in mice; both families are responsible to shuttle lipophilic biomolecules [9,11,12].
Furthermore, we had already noted this in the Results section 3.5. as well as in other parts of the manuscript.
The probasin coding sequence seems to share similar but not always identically sized exons of the lipocalin family, suggesting close relationship, potentially indicating that probasin might have emerged as lipocalin outlier gene.
- Please discuss how the probasin gene can affect the evolution of rodent?
Response 7:The following paragraph has been added at the end of the discussion section briefly discussing its potential role.
“Probasin has a close structural relationship to other lipocalins such as major urinary proteins and odorant-binding proteins, which are both involved in the shuttling of pheromones [12]. Combinations of these proteins are supposed to potentially have a signature role for different species [12,34]. The addition of probasin to this ‘cocktail’, which is produced by the prostate and, hence, be added to seminal fluid, might contribute to a signature blend of odorants shuttled by MUPs and OBPs. We need to await the generation of probasin-deficient mice to assess its potential beneficial evolutionary role.”
- Only use RT-PCR in bioinformatics analysis were considered simple work. Please find some other data with transcriptomic, proteomic and epigenomic data, to make more comprehensively understand the role and regulation mechanism of probasin gene in different biological processes
Response 8: Johnson and co-workers published in their probasin analysis that due to regulatory motifs in the probasin promoter, protein expression is severed once animals are castrated. This emerged as the only study actively analyzing protein expression and regulation in vivo. As this reviewer knows, only protein expression tells something about the role and function of protein coding genes. Changes in transcription levels, however, might indicate a role, which needs to be determined. Given that we still have to await the generation of probasin deficient mice, its in vivo role deduced by studying its deficiency remains speculative.
Exploring transcriptomic databases is, if at all, only worthwhile in mice, since probasin is absent in humans, or more specifically outside the Muroidea superfamily, which has been the key message of our manuscript.
With all due respect further speculative assessment of probasin on a transcriptomic level is entirely out of the scope of our manuscript. We agree that further studies are necessary, we believe however that this needs to be part of future studies exploring probasin deficient mice.
It is a pity that our time consuming efforts to screen a large amount of data sets is considered “simple work”. Given that most entries outside of human, mouse and rat in NCBI databases are merely predicted and not carefully evaluated, we want to raise awareness that our body of work has proven many times that we do not have to take for granted what is published in public databases, or at least have to take it with a grain of salt.
We added to our discussion section a brief statement (see above) suggestion that probasin might add to the cocktail of signature odors in seminal fluid in the last paragraph.
Reviewer 3 Report
Comments and Suggestions for Authors
The article titled "Comprehensive Analysis of the Rodent-Specific Probasin Gene Reveals Its Evolutionary Origin in the Pseudoautosomal Region and Provides Novel Insights into Rodent Phylogeny" examines the probasin gene’s evolutionary significance within the Muroidea superfamily. This review provides concise observations and suggestions for refinement.
The abstract is clear but could emphasize the phylogenetic importance of the findings more effectively. The introduction sets the context well but would benefit from a stronger link to broader evolutionary questions. The methodology is detailed and appropriate, though clarification of the selection criteria for rodent genomes would enhance transparency.
The results reveal significant findings, including gene conservation, GC content differences, and recombination events. Figures, such as Figures 1 and 5, could be simplified for improved clarity. The discussion aligns with the study’s objectives, proposing hypotheses about probasin’s role. A deeper exploration of functional implications and broader evolutionary impacts would strengthen this section.
The manuscript is precise, though reducing technical jargon would improve accessibility. Simplifying visuals and better contextualizing findings for evolutionary studies beyond rodents would enhance the work’s relevance.
The study has limitations. Its focus on rodent genomes within Muroidea leaves questions about broader applicability to other mammals. The absence of experimental validation for the proposed functional roles of probasin proteins confines the conclusions to theoretical frameworks. Future research integrating functional assays and comparative analyses with non-Muroidea species could address these gaps.
Author Response
We would like to thank Reviewer 3 for his/her/their criticism, and the comments raised. Please, find below a point-to-point reply to the comments. We would like to point out, that some of the criticism might have already been addressed in context with the response to the comments/criticism raised by Reviewer 1 and Reviewer 2.
We hope that Reviewer 3 finds our corrections and improvements appropriate to eventually granting his/her/their approval for publication in Biology.
Thank you and kind regards,
Stephan Maxeiner on behave of all authors
Point-to-point reply
Comment 1: The article titled "Comprehensive Analysis of the Rodent-Specific Probasin Gene Reveals Its Evolutionary Origin in the Pseudoautosomal Region and Provides Novel Insights into Rodent Phylogeny" examines the probasin gene’s evolutionary significance within the Muroidea superfamily. This review provides concise observations and suggestions for refinement.
Response 1: Thank you, we appreciate this kind comment.
Comment 2: The abstract is clear but could emphasize the phylogenetic importance of the findings more effectively.
Response 2: We have changed the middle section of the abstract and also added the simple summary, which is an essential part of Biology articles.
Comment 3: The introduction sets the context well but would benefit from a stronger link to broader evolutionary questions.
Response 3: Upon suggestion by Reviewer 1, we have revised the introduction. We hope that the importance and problems in studying rodents for human diseases was addressed, furthermore, we point out that the study of the rodent lineage is important given that they comprise 40% of all mammal species.
Comment 4: The methodology is detailed and appropriate, though clarification of the selection criteria for rodent genomes would enhance transparency.
Response 4: We have revised the methods section and added a flowchart as has been suggested by Reviewer 2. We commented on how the number of 61 genomes has come together. We hope that this adds transparency.
Comment 5: The results reveal significant findings, including gene conservation, GC content differences, and recombination events. Figures, such as Figures 1 and 5, could be simplified for improved clarity.
Response 5: We have split up the original Figure 1 into two separate figures, now new Figure 2 and new Figure 5. Given that the content of Figure 5 in fact was presented after the sequence analysis in previous Figure 3 and Figure 4, we believe that this adds overall clarity and helps us to drive our message home. Previous Figure 5, now new Figure 7, which included our findings regarding the genomic localization, was “decluttered” in a fashion that the reading direction of each gene was removed (previously indicated by an arrow), information that was not necessary to convey our thoughts, and the name of the genes have been inserted into the colored boxes.
Comment 6: The discussion aligns with the study’s objectives, proposing hypotheses about probasin’s role. A deeper exploration of functional implications and broader evolutionary impacts would strengthen this section.
Response 6: We added a section at the end of the Discussion, in which we speculate about the impact of probasin arising in rodents. We are reluctant to further speculate, given that yet probasin deficient mice await to be generated. In fact, all the work helping to unravel the functions of either lipocalins, major urinary proteins or odorant binding proteins in mice that has been done needs to be repeated with probasin. Its limited expression to prostate tissue and missing link to human pathologies let researchers thus far refrain from being invested into further exploring probasin. This, however, is a pity given its presence in many rodent species. This is why we felt propelled to start this comprehensive analysis yielding an engaging primer for further studying the role of this outlier lipocalin.
Comment 7: The manuscript is precise, though reducing technical jargon would improve accessibility. Simplifying visuals and better contextualizing findings for evolutionary studies beyond rodents would enhance the work’s relevance.
Response 7: As stated above, we revised a fseveral figures, which is also partially based on suggestions by both other reviewers. Previous Figure 1 combined gene organization and nucleotide composition. We split this figure up into two separate figures, now Figure 2 and Figure 5. During the revision we realized that the content presented in Figure 5 is referred to at a later section in the Results part. As suggested by Reviewer 2, we added a flowchart depicting the process of fetching probasin sequences from the NCBI database. This figure proceeds all other figures and has therefore been introduced as Figure 1. Original Figure 4, which depicts the phylogenetic relationship has been modified, now highlighting branches with bootstrap values over 70%. This makes Supplementary Figure 1 obsolete, hence referring to former Suppl. Figure 2 now as Suppl. Figure 1. Finally, original Figure 5 summarizing the rearrangement during evolution regarding the localization of the probasin gene, has been modified. The information about the reading frames of each gene is regarding its message irrelevant. Therefore, the white arrows in the colored boxes have been removed and the gene names have been inserted. We believe this helped to declutter the previous figure. The revised figure appears now as Figure 7.
We believe that our representation of genomic relationships and names for genes, etc. is comparable to other publications in this field. To address the use of jargon, we decided to add after their respective first mentioning in the text, the full name in brackets next to the abbreviated gene name. Additionally, the protein motifs in section 3.5. have been specified. The mentioning of splice donor and accdeptor sides was clarified. The letters representing the nucelotides were mentioned in full in the Abbreviation section.
Comment 8: The study has limitations. Its focus on rodent genomes within Muroidea leaves questions about broader applicability to other mammals. The absence of experimental validation for the proposed functional roles of probasin proteins confines the conclusions to theoretical frameworks. Future research integrating functional assays and comparative analyses with non-Muroidea species could address these gaps.
Response 8: We agree, that this is theoretical framework, but we hope that this comprehensive analysis of probasin genes in rodents represents an engaging primer for further studies.
Round 2
Reviewer 2 Report
Comments and Suggestions for Authors
The quality of manuscript have been improve a lot. Now it can be accept to publish in our journal.
Author Response
We would like to thank Reviewer 2 for her/his/their feedback, and we appreciate their decision.
Stephan Maxeiner on behalf of all authors.